# Translation of Multidimensional Health Locus of Control Scales, Form C in Patients with Headache

**DOI:** 10.3390/jcm13144239

**Published:** 2024-07-20

**Authors:** Luise Bartsch, Nadja Fiebig, Sebastian Strauß, Anselm Angermaier, Craig A. Smith, Uwe Reuter, Robert Fleischmann

**Affiliations:** 1Department of Neurology, University Medicine Greifswald, 17475 Greifswald, Germany; luise.bartsch@med.uni-greifswald.de (L.B.);; 2Department of Psychology and Human Development, Peabody College, Vanderbilt University, Nashville, TN 37203, USA

**Keywords:** migraine, headache, locus of control, MHLC-C

## Abstract

The MHLC-C is a condition-specific instrument measuring the internal and external loci of control beliefs, adaptable to various health conditions. Translated into Swedish and Chinese, this study aims to translate the MHLC-C into German using the FACID-Method. The English version is validated and reliable; the German version requires these validation steps.

## 1. Introduction

Headaches, particularly migraines, significantly impact patients’ daily activities, professional responsibilities, and social lives [1]. Migraines are associated with mood changes, psychosocial difficulties, and a reduced quality of life. Enhancing health-promoting behaviors can improve the condition and patients’ coping strategies. The health locus of control (HLoC) construct, derived from Rotter’s social learning theory and developed by Wallston, explains health-related behaviors and influences a person’s health status [2]. HLoC refers to beliefs about control over one’s health, either internal or external. Those with a high internal LoC believe their actions primarily determine their health outcomes, leading them to adapt their behavior, take an active interest in their health, and exhibit fewer depressive disorders. In contrast, those with a high external LoC believe their health is largely determined by physicians, other people, or chance. [3]. Rotter emphasized specific expectations over generalized ones to predict behaviors in specific situations better. Wallston addressed this by developing the MHLC-C, which measures internal and external control beliefs with four subscales: ‘internal’, ‘chance’, ‘other people’, and ‘doctors’. This tool explains intra- and interindividual differences in health behaviors and individualizes interventions.

Although the MHLC-C has been applied to various conditions, its impact on headache management is underexplored. Consequently, translating and validating the MHLC-C for German-speaking populations is crucial for improving headache-related health outcomes in these regions. This study aimed to provide a state-of-the-state translation of the MHLC-C for public use in German-speaking countries. This would allow for evaluating and possibly optimizing multimodal treatment programs in consideration of the construct of LoC.

## 2. Materials and Methods


**
*Translation and cross-cultural adaptation*
**


With the increase in multinational and multicultural research projects, there is a growing need to adapt health status measures for use in languages other than the source language [4,5]. Cross-cultural adaptation must be systematic to establish equivalence between the source and target versions of questionnaires. The FACIT translation methodology is well-established for this purpose. [6].


**
*Translation using the FACIT methodology*
**


The FACIT methodology involves a forward and backward translation by a multidisciplinary team, followed by several steps to ensure consistency with the source, culminating in a cognitive debriefing. The translation process is summarized in Figure 1. Three native German speakers (including one with a professional qualification in English language and literature studies, one neurologist, and one psychologist) conducted a forward translation from English to German blinded to each other’s translations. The diverse backgrounds of the translators ensured a comprehensive translation that captured both linguistic nuances and medical context. The three versions were then reviewed by all three and a concerted translation was established. A professional translator, who was also an English native speaker, then performed a backward translation from German into English. Comments were discussed among the three German translators to reconcile the translations. Discrepancies were resolved through consensus, ensuring that the final items closely matched the original content. Cognitive debriefing involved ten individuals who rated the clarity and comprehensibility of the German MHLC-C items. This step was crucial for identifying and correcting any issues with translation accuracy and cultural relevance. With regard to the closest possible proximity to the original content, problematic items were modified. Difficulties in translating items were rated on a 5-point Likert scale (1 = very easy, 5 = very difficult).


**
*Translation of the questionnaire*
**


The original version includes eighteen questions (see Appendix A for original items). In general, the MHLC-C questionnaire is a condition-specific locus of control scale, which can be used when studying people with a health- or medical-related condition. Unlike Forms A and B, which measure generalized health locus of control, Form C includes three specific external subscales: ‘doctors’, ‘chance’, and ‘other people’. Wallston suggested replacing the word “condition” in each item with the specific condition the subjects have [8]. For our study, ‘condition’ was replaced with ‘headache’ to specifically address the needs of individuals suffering from this condition, reflecting the focus of our research.

## 3. Results

The FACIT methodology was performed according to protocol. The particular working steps can be seen in Figure 1. For a better overview, we divided the introduction into four parts. The finalized translation can be found in Appendix A. During cognitive debriefing, most items (1, 4, 7, 8, 9, 11, 12, 15, 16, and 17) were rated as easy to understand by the participants. Items 2, 3, 5, 10, 13, and 14 were rated as moderately challenging, and item 18 was rated as difficult to translate. These ratings were based on participants’ feedback on clarity and comprehensibility. Only limited changes were made in order to remain as close as possible to the original wording.

Specific adjustments included replacing ‘item’ with ‘Aussage’ for better comprehension. Following a cognitive debriefing, the response options were revised for clarity, such as replacing ‘mäßig’ with ‘überwiegend’. Item 2 was modified to ‘… geschieht, was geschieht’ due to poor understanding during the debriefing, despite no back-translation issues. Similarly, item 4 was changed to ‘dann habe ich wahrscheinlich weniger Probleme mit meinen Kopfschmerzen’ for better clarity.

The cognitive debriefing revealed issues with some translated items, so they were modified for better intelligibility. In item 6, “selbst” was removed, in item 8, we used “schief läuft” instead of “falsch läuft”, and in item 13, “die” was removed. In item 14, we removed the negation that was mistakenly inserted in the translation process. In the cognitive debriefing, it was noted that some items of the subscale “Chance” sounded very similar (item 9, 11, and 16), but no changes were made. Refer to Appendix A for the original items and the German translation.

## 4. Discussion

The MHLC-C is a versatile, condition-specific locus of control scale, adaptable for various medical and health-related conditions, including rheumatoid arthritis, type 2 diabetes, and HIV [9,10,11]. The availability of a German version is significant, as it allows for culturally relevant research and application in German-speaking populations. While the English version of the MHLC-C is validated and reliable, the German version still requires psychometric validation to ensure its reliability and validity within the German-speaking context.

### Locus of Control in Headache Management and Research

Locus of control (LoC) is crucial in the effectiveness of non-pharmacological treatments as part of multimodal headache management. Internal control beliefs have been shown to affect pain reduction positively. For instance, in multidisciplinary inpatient treatments with chronic pain, internal health-related LoC has demonstrated predictive value for reducing pain intensity. [12]. Further research by Wallston found that severely ill patients with a high external health control orientation benefited more from behavioral interventions [13].

To test the hypothesis that patients with moderate values on the subscale “internal” and high values on the subscale “doctors” benefit the most from multidimensional therapy, this research instrument is required. For research on predictors for success of multimodal headache treatment, this could mean that a moderate internal locus of control orientation and enhancing or lowering specific external locus of control beliefs could lead to fewer headache days, a lower headache frequency, higher medication adherence, and greater application of coping strategies [14,15].

## Figures and Tables

**Figure 1 jcm-13-04239-f001:**
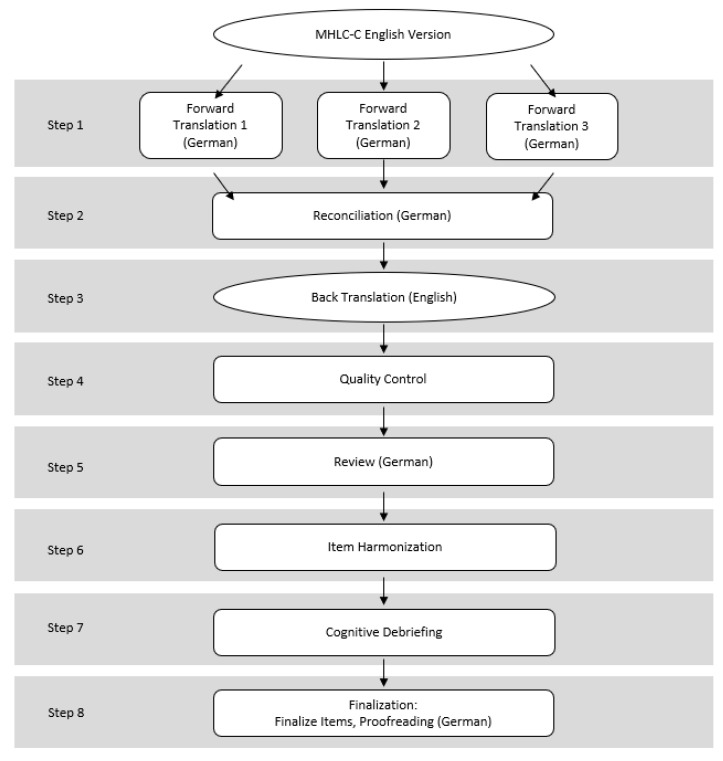
A modified flowchart of the Functional Assessment of Chronic Illness Therapy (FACIT) translation methodology by Eremenco et al. [7].

## Data Availability

The datasets used and/or analyzed during the current study are available from the corresponding author on reasonable request.

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
