# Peer review of "Translation of Multidimensional Health Locus of Control Scales, Form C in Patients with Headache"

_jcm, 2024, doi:10.3390/jcm13144239_

Round 1
Reviewer 1 Report
Comments and Suggestions for Authors
This is a review on multidimensional health locus of control scales, translation of Form C into German language. Authors presented background of multidimensional health locus of control scales in the introduction. I would recommend to describe specific challenges to use MHLC-C and how it would benefit the population. Methodology was explained clearly and flowchart was good. Questionnaire was translated into German, I would recommend to send this review to a physician or neurologist who speaks German.
Reviewer 2 Report
Comments and Suggestions for Authors
The topic of locus of control is a key factor in the management of health disorders and particularly relevant to pain. That it has not been used in headache makes it even more so. My only concern here is that for an English language paper the nuances of translation to and back from German cannot be appreciated unless one has knowledge of both languages. Therefore, the question is whether a German language journal would be more suitable or whether there is a way of explaining the nuances between the languages for an English language journal.
Reviewer 3 Report
Comments and Suggestions for Authors
Abstract
1. Clarity and Conciseness: The abstract is clear but can be made more concise.
- Suggested Change: "The MHLC-C is a condition-specific instrument measuring internal and external locus of control beliefs, adaptable to various health conditions. Translated into Swedish and Chinese, this study aims to translate the MHLC-C into German using the FACID-Method. The English version is validated and reliable; the German version requires these validation steps."
2. Completion: Ensure the abstract succinctly covers all necessary aspects: the tool, its purpose, and the aim of the study.
- Suggested Addition: Briefly mention the significance of the German translation.
Introduction
3. Flow and Structure: The introduction is well-structured but could benefit from more fluid transitions between ideas.
- Suggested Change: "Headaches, particularly migraines, significantly impact patients' daily activities, professional responsibilities, and social lives. Migraines are associated with mood changes, psychosocial difficulties, and reduced quality of life. Enhancing health-promoting behaviors can improve the condition and patients' coping strategies."
4. Explanation of Concepts: The explanation of HLoC is good but can be slightly refined for better readability.
- Suggested Change: "The health locus of control (HLoC) construct, derived from Rotter's social learning theory and developed by Wallston, explains health-related behaviors and influences a person’s health status. HLoC refers to beliefs about control over one’s health, either internal or external. Those with high internal LoC believe their actions primarily determine their health outcomes, leading them to adapt their behavior, take an active interest in their health, and exhibit fewer depressive disorders. In contrast, those with high external LoC believe their health is largely determined by physicians, other people, or chance."
5. Transition to Specific Instrument: Discussing the MHLC-C can be smoother.
- Suggested Change: "Rotter emphasized specific expectations over generalized ones to predict behavior in specific situations better. Wallston addressed this by developing the MHLC-C, which measures internal and external control beliefs with four subscales: 'internal,' 'chance,' 'other people,' and 'doctors.' This tool explains intra- and interindividual differences in health behaviors and individualizes interventions."
6. Relevance to Headache Management: The connection to headache management is clear but can be more explicitly stated.
- Suggested Change: "Although the MHLC-C has been applied to various conditions, its impact on headache management is underexplored. Consequently, translating and validating the MHLC-C for German-speaking populations is crucial for improving headache-related health outcomes in these regions."
Materials and Methods
7. Introduction to Translation and Cross-Cultural Adaptation:
- Clarity: The introduction to the need for cross-cultural adaptation could be more concise.
- Suggested Change: "With the increase in multinational and multicultural research projects, there is a growing need to adapt health status measures for use in languages other than the source language. Cross-cultural adaptation must be systematic to establish equivalence between the source and target versions of questionnaires. The FACIT translation methodology is well-established for this purpose."
8. Translation Using the FACIT Methodology:
- Consistency in Terminology: Ensure consistent use of terms like "forward translation" and "backward translation."
- Suggested Change: "The FACIT methodology involves a forward and backward translation by a multidisciplinary team, followed by several steps to ensure consistency with the source, culminating in a cognitive debriefing."
9. Forward Translation Process:
- Detail on Participants: The qualifications of the forward translators are well-stated, but specifying the importance of their backgrounds might be beneficial.
- Suggested Addition: "The diverse backgrounds of the translators ensure a comprehensive translation that captures both linguistic nuances and medical context."
10. Back-Translation and Reconciliation:
- Clarity on Reconciliation: The reconciliation process should emphasize how discrepancies were resolved.
- Suggested Change: "Comments were discussed among the three German translators to reconcile the translations. Discrepancies were resolved through consensus, ensuring the final items closely matched the original content."
11. Cognitive Debriefing:
- Detail on Cognitive Debriefing: More information on the cognitive debriefing process and its importance would be helpful.
- Suggested Addition: "Cognitive debriefing involved ten individuals who rated the clarity and comprehensibility of the German MHLC-C items. This step is crucial for identifying and correcting any issues with translation accuracy and cultural relevance."
12. Translation of the Questionnaire:
- Consistency in Describing Forms: When describing the differences between Forms A, B, and C, ensure clarity and consistency.
- Suggested Change: "Unlike Forms A and B, which measure generalized health locus of control, Form C includes three specific external subscales: 'doctors,' 'chance,' and 'other people.'"
- Clarify the Adaptation for the Study: Make it clear why "headache" was chosen as the specific condition.
- Suggested Addition: "For our study, 'condition' was replaced with 'headache' to specifically address the needs of individuals suffering from this condition, reflecting the focus of our research."
Results
13. Detail on Cognitive Debriefing:
- Understanding Ratings: Provide more context on how the ratings (easy, moderately challenging, difficult) were determined during cognitive debriefing.
- Suggested Change: "During cognitive debriefing, most items (1, 4, 7, 8, 9, 11, 12, 15, 16, 17) were rated as easy to understand by the participants. Items 2, 3, 5, 10, 13, and 14 were rated moderately challenging, and item 18 was rated difficult to translate. These ratings were based on participants' feedback on clarity and comprehensibility."
14. Specific Item Adjustments:
- Clarify Changes and Rationale: Explain why specific items were modified and how these changes improved the translation.
- Suggested Change: "Specific adjustments included replacing 'item' with 'Aussage' for better comprehension. Following a cognitive debriefing, response options were revised for clarity, such as replacing 'mäßig' with 'überwiegend.' Item 2 was modified to '… geschieht, was geschieht' due to poor understanding during debriefing, despite no back-translation issues. Similarly, item 4 was changed to 'dann habe ich wahrscheinlich weniger Probleme mit meinen Kopfschmerzen' for better clarity."
Discussion
15. Context and Importance:
- Importance of Translation: Emphasize the significance of having a validated German version of the MHLC-C.
- Suggested Change: "The MHLC-C is a versatile condition-specific locus of control scale, adaptable for various medical and health-related conditions, including rheumatoid arthritis and HIV. The availability of a German version is significant as it allows for culturally relevant research and application in German-speaking populations."
16. Pending Validation Steps:
- Detail on Validation: Clearly state what validation steps are pending for the German version.
- Suggested Change: "While the English version of the MHLC-C is validated and reliable, the German version still requires psychometric validation to ensure its reliability and validity within the German-speaking context."
17. Relevance to Headache Management:
- Connection to Headache Management: Strengthen the connection between locus of control and its impact on headache management.
- Suggested Change: "Locus of control (LoC) is crucial in the effectiveness of non-pharmacological treatments as part of multimodal headache management. Internal control beliefs have been shown to affect pain reduction positively. For instance, in multidisciplinary inpatient treatments for chronic pain, internal health-related LoC has demonstrated predictive value for reducing pain intensity."
18. References
Most of the references were very old, so incorporate recent ones.
References not followed. Kindly rewrite the references as per the Vancouver style of writing.
Round 2
Reviewer 2 Report
Comments and Suggestions for Authors
My comments really is as before that the nuances of the German language in this paper make it difficult to assess the results as an English speaker. Thus, unless there can be a way of explaining this in English the paper may be better in a German language journal.
